# Health Promotion and Disease Prevention in the Elderly: The Perspective of Nursing Students

**DOI:** 10.3390/jpm12020306

**Published:** 2022-02-18

**Authors:** Rogério Ferreira, Cristina Lavareda Baixinho, Óscar Ramos Ferreira, Ana Clara Nunes, Teresa Mestre, Luís Sousa

**Affiliations:** 1Comprehensive Health Research Centre, 1150-082 Lisboa, Portugal; ferrinho.ferreira@ipbeja.pt (R.F.); lmms@uevora.pt (L.S.); 2Polytechnic Institute of Beja, 7800-000 Beja, Portugal; ana.nunes@ipbeja.pt (A.C.N.); teresa.mestre@ipbeja.pt (T.M.); 3Center for Innovative Care and Health Technology (ciTechCare), Polytechnic of Leiria, 2410-541 Leiria, Portugal; 4Nursing Research, Innovation and Development Centre of Lisbon (CIDNUR), Nursing School of Lisbon, 1600-190 Lisboa, Portugal; oferreira@esel.pt; 5S. João de Deus Higher School of Nursing, University of Évora, 7000-811 Évora, Portugal

**Keywords:** aged, disease prevention, health promotion, nursing education, nursing students

## Abstract

Health promotion and disease prevention are closely linked to health literacy. Therefore, intervention to increase individuals’ knowledge is essential if action is to be taken to promote a healthy lifestyle with support from health professionals for decision making on choices leading to behavioral change. Taking into account the growing aging population, nurses and nursing students have to develop interventions to promote health and prevent disease in these people, in order to keep them healthy and with quality of life. This study aims to understand how nursing students’ experiences in a clinical teaching context contributed to the development of their competencies in the promotion of health and prevention of disease in the elderly. Method: Qualitative, exploratory, and descriptive study carried out with ten students about to finish a graduate nursing course in a higher education institution in the South of Portugal. This study was carried out through narratives, one of the most common data collection procedures in social and health investigations. The content analysis technique, more specifically the thematic categorical analysis, was used for data analysis. The study received authorization from the Ethics Committee of the institution where it took place. Results: Three categories were found: “Strategies to promote health and prevent disease in the elderly”, “Health improvements from the implementation of the strategies to promote health and prevent disease in the elderly”, and “The impact your participation in these strategies to promote health and prevent disease in the elderly had on your formative process”. Conclusion: The students developed competencies during their clinical teaching experiences through the implementation of strategies of health promotion and disease prevention adapted to/focused on the needs of the elderly.

## 1. Introduction

Portugal has one of the oldest populations in the world. Nonetheless, projections show that the Portuguese population may go from 2.1 million elders in 2015 to 2.8 million people in 2080, and the aging index may duplicate, from 147 to 317 elderly people per 100 young people [1]. Demographic aging has been a target of study for authors and students from many fields of knowledge, and it is not uncommon to hear about the “problem of demographic aging” [2]. The aging of the Portuguese population and the increase in mean life expectation suggests that there are improvements in mortality rates in adults and elderlies [1,3]. Therefore, this is not a problem per se but also an advantage, as it mostly reflects improvements in the living conditions and health policies of the so-called developed societies. However, it is undeniable that due to the aging process, i.e., biological aging, gradual changes occur in the human organism’s ability to adapt. As a result, people become increasingly likely to acquire acute and chronic diseases and disabilities, with repercussions on their quality of life. This context is coupled with the emergence of new needs that force us to characterize the phenomenon and rethink the roles and rights of the elder. The aging of society is unavoidable, but as it advances, dependence, vulnerability, and weakness also grow [4,5]. These conditions are concerning, and we must carry out interventions in regard to them.

Although the mean life expectancy in Portugal is higher than the mean of the other Organization for Economic Co-operation and Development (OECD) countries [6], the index “number of healthy years of life after 65 years old” is one of the lowest recorded ones. Despite living longer, we live with more comorbidities during our last years of life, which include diabetes, cardiovascular diseases, respiratory diseases, obesity, and cancer [6]. This is why health promotion and disease prevention in the elderly should be promoted to foment and maximize healthy, active, and successful aging [7].

Health promotion and disease prevention are intimately tied with knowledge and, therefore, with health literacy, among other variables. Health promotion treats individuals as active agents, with the power and capacity to control and make decisions about their own health [8]. Therefore, increasing the knowledge of individuals is essential for more assertive and health-promoting measures to be taken. Generally, health education is the chief strategy to train people, groups, or communities. According to studies about this topic, factors that influence an individual’s health literacy are static, but knowledge is dynamic. Therefore, knowledge is a modifiable construct that can be changed by interventions appropriate to health literacy [9]. Hence, education in health intends to increase the knowledge of individuals, providing them with tools that allow them to learn better, increasing their knowledge, and developing competencies that favor their own health and that of the community around them [10].

In the scope of their clinical classes, students of the graduate nursing course “learn, within a team which has direct contact with an individual and/or collectivity that may be in a good state of health or ill, to plan, provide, and evaluate the global nursing care required, based on the knowledge and competences acquired” (Norm 2005/36/CE from the European Parliament and from the Council, 7 September 2005, No. 5, article 3°) [11]. Clinical classes are articulated with the other disciplines as it mobilizes knowledge and abilities, consolidating and complementing what was learned—in particular, what was seen in previous clinical teaching classes—to achieve improved health results, ensuring that the learning processes are increasingly complex. The clinical classes in Community Health Nursing of the nursing graduation course at the Instituto Politécnico de Beja (IPBeja), in particular, aim to develop competencies that allow the student to provide nursing care to individuals, including their family and community, leading them to become increasingly thoughtful citizens that can bring improvements in health. The clinical classes take place in Primary Health Care Units, preferably at the family healthcare unit, where students are integrated into the activities of the team/unit, recognizing needs and providing care to persons, their families, and to the community, in the scope of the Primary Health Care. Students also develop, in groups, a community intervention project, using the health-planning methodology to deal with the needs of the unit/community where they carry out their clinical classes [12].

We believe that the results of this study, involving students who had the opportunity to live rich and diversified experiences in clinical settings of community health care delivery and focused on health promotion and disease prevention in the elderly, may be an important contribution to reflection and analysis on health promotion and disease prevention practices in the elderly from the perspective of nursing students in learning environments.

As a result, this study included students who had full and diverse experiences, in a clinical context, of providing health to a community, focusing on the promotion of health and prevention of disease in the elderly. This study aims to understand how nursing students’ experiences in a clinical teaching context contributed to the development of their competencies in the promotion of health and prevention of disease in the elderly. We believe that the results of this study may be an important contribution to the reflection on this issue and the relevance of learning experiences in clinical settings, aiming at the development of health promotion and disease prevention skills in older people.

## 2. Background

The growing number of community-dwelling elderly and the increased risks of adverse health events that accompany aging call for health promotion interventions [13].

The maintenance of the elderly in their social context fits in with the strategies for an active, participatory, healthy aging and is an illustration of true centered care with a holistic and integrative vision, emphasizing self-determination and valorization of individual characteristics, expectations, and potentialities and assigning them an active role as decision-makers as agents of their own lives [8,14,15,16,17]. For this to happen, health promotion and disease prevention in this population should involve the adoption of healthy lifestyles and the inclusion of strategies promoting health literacy [9,10], contributing to their learning and development of skills that promote their health. The strengthening of elderly knowledge is primordial for taking more assertive and healthier measures and options. Health Education is the “mother strategy” for the empowerment of people, groups, or communities [8].

This preventive approach is a strategy at the primary health care level and requires significant community participation, with the involvement of multiple partners through mechanisms to improve people’s literacy level and the creation of environments conducive to promoting and protecting elderly health [7,8,9,13,14,15,16,17].

Although health promotion is seen as a major activity of health professionals, it should include the coordination between health institutions and community partners through programs conducive to healthy and successful aging promoting social interaction and mental health, promotion of physical activity, promotion of healthy eating habits, prevention and self-monitoring of chronic diseases, promotion of a medication regimen adjusted to the needs, promotion of safety measures [18], and accident prevention [19].

The provision of proximity health care in the social context where the elderly lives should be a response that promotes healthy, active, and successful aging, with the quality of life that it implies [18]. The principle of aging in place advocates that the elderly should remain in their own place, in their own homes, and in the social context that contributes to their autonomy, according to their preferences [8]. The difficulties and limitations that arise with the advancing age can be minimized with social and health responses, which are compatible with policies for caring elderly at home and are promoters of active and successful aging [8,14,15].

When addressing the subject of training or education aimed preferably at the elderly, we inevitably talk about gerontagogy, an expression advocated by Lemieux in 1997, which in general terms is defined as an interdisciplinary educational discipline that has as its object the study of the elderly in an educational situation [20]. This hybrid scope that combines two specialties, i.e., pedagogy and gerontology, is a category that bets on the renewal of the ways of thinking about education and aging [8]. Gerontagogy constitutes, above all, a paradigm, a concrete way of betting on the education of elderlies with dependencies or not [20].

In their role as “educators” of this population, nurses should always bear in mind that this is a special pedagogy because it is addressed to a population with specific characteristics, but they should also rethink and adjust the teaching methods, which should be as participatory as possible, with the elderly playing an active and leading role [8,13,14,15,16]. We agree that in parallel with the increase in the aging population and a change in the disease spectrum, the nurse’s role is shifting from the traditionally following physicians’ orders to a more expanded role with responsibility for preventive care [8]. Nurses also pay more attention to health promotion advice, lifestyle counseling, educational programs, and the provision of early interventions to prevent exacerbation or complications for persons living with diseases [8,9,10,11,13].

These actions should be appropriate to the needs of the elderly and are justified taking into account the existence of risk factors. They should contribute to their ability to manage their lives with functional independence and have an optimistic perception of life associated with their participation in social and leisure activities [8,9,10,16]. Health gain’s studies resulting from the implementation of health promotion programs highlight the increase in functional capacity, quality of life, and social participation, the improvement in knowledge and awareness about lifestyles and their impact on elderly health, the gains in terms of mental health, and the decrease in the number of medical consultations and hospitalizations [14,15,18].

In view of the above, and with this perspective that the promotion of active aging, preserving the elderly’s abilities, is one of the aims of the intervention of health professionals, particularly nurses [13,14,15], it is important that nursing students acquire not only knowledge about health literacy [9,10], health education techniques, and behavioral change but also develop skills to facilitate elderly’s access to health information and support decision making, thus achieving better health outcomes [8,16].

## 3. Materials and Methods

This study was qualitative, exploratory, and descriptive, focused on the relevance of the care provided by the nurse and aiming to promote health and prevent disease in the elderly, from the perspective of nursing students going through the learning process.

The participants are in the last period of the nursing graduation course of a public higher education institution in the South of Portugal. They successfully finished the discipline Nursing Clinical Classes in Community Health. The sample was intentional (non-probabilistic), selected by the investigators. It involved students with rich and diverse experiences in a clinical context of the provision of health care to the community, focusing on the promotion of health and the prevention of disease in the elderly. Inclusion criteria adopted were as follows: being a student in the 4th year of the nursing graduation course of a public higher education institution in the South of Portugal, having been approved in the discipline Nursing Clinical Classes in Community Health, having been involved in practices of health promotion and disease prevention in the elderly, and being willing to participate in the study. There were no exclusion criteria.

Data were collected using narratives since in social and health investigations, this is one of the most used procedures to gather information on the meaning of experiences [21] of students in clinical contexts in regard to the provision of nursing care to the elderly and their families. A script to guide the narratives was created and submitted to the judges to guarantee the content validity of the instrument. The main concern of this project was to guarantee that the guiding questions of the narrative were in accordance with the objectives defined and allowed for a reorganization of the experiences of care to the elderly in a coherent and significant way, giving meaning to the experience and enabling its (re)construction as a process integrated into the context of clinical practice [21]. The script aimed to provoke reflections and analysis about the practices of health promotion and disease prevention in the elderly from the perspective of nursing students in their learning process. When writing/narrating about the care practices, students reconstructed and re-organized their experiences, giving meaning to the events that supported the narrative in a way consistent with their current understanding. This means that its use was based on the assumption of valuing reflection on clinical practice as a strategy that ensures the reconstruction of knowledge based on the reflected knowledge. The script included three guiding questions:− In what strategies to promote health and prevent disease in the elderly were you involved?− What health benefits emerged from the implementation of these strategies to promote health and prevent disease in the elderly?− What was the impact of these strategies in your education process?

All necessary ethical precepts that must guide the elaboration of a study such as this were followed in any contact with the higher education institution and its students. On the institutional level, the study with Process No. 24/2021 received a favorable evaluation on 8 July 2021 from the Ethics Commission of the Instituto Politécnico de Beja. The participating students were assured that their participation in the study would be entirely voluntary and that they could abandon it at any time, with no need to justify their decision or any future consequences. Their anonymity was guaranteed, as well as the confidentiality of their data since professional secrecy is both an obligation and a duty. In addition, they signed the free and informed consent form.

Participants were contacted by the main investigator, who presented the investigation project and explained the goal of the study, its objectives, and the importance of participating. It also involved the expression of availability to clarify doubts and ensure understanding of the guiding questions of the reflection. The study was carried out through e-mail.

The students elaborated their narratives in response to the guiding questions. These were sent via e-mail to the main researcher in a pdf file.

The main researcher was responsible for managing the data collected, ensuring the anonymity and confidentiality of the data. All information that could identify the participants was restricted, and each participant received a name. The names of the participants will be replaced by identification numbers (P1, P2, P3…) in all records and publications. In the topics (content units) presented in the results of the study, measures to protect the identity of the students were taken. Data protection went from the selection of the participants to the collection, analysis, and dissemination of the results of the study.

The information collected and submitted to analysis, in addition to the free and informed consent form documents, was stored in a drive owned by the main researcher responsible for this study and will remain stored for five years. The data were encrypted, including raw data, to prevent access to it by unauthorized third parties. Other people will not have access to this information.

The narratives, more than a document, are a primary source that integrates the reflection of these students [21]. They were analyzed using the content analysis technique, more specifically, the thematic category analysis [22]. Its choice is related to the fact that it is a set of communication analysis techniques, in which systematic and objective procedures to describe the content of messages are used to obtain indicators that allow the inference of knowledge concerning the conditions of production of the messages. In the first stage, the texts were skimmed to verify whether the information collected was in accordance with the objective of the study. The narratives formed the corpus of the analysis, that is, the material produced for the investigation to be analyzed [22]. The second stage was the coding of the data, with the definition of thematic units and context units. Thematic units, according to Bardin [22], are statements about a subject to which a vast range of unique formulations may be associated. Contextual units—the “comprehension units” used to understand the exact meaning of each thematic unit—was the response of each participant to the guiding questions for the narrative. In addition to defining coding units, categories and indicators were defined. In the third stage, the results were treated through inference, attributing meanings to the qualitative analysis of the categories.

For the thematic category analysis, the text was separated into thematic units (themes) and categories, according to analogical regroupings [14]. In this process, semantic categorization criteria were used.

Three categories were found in this study: “Strategies to promote health and prevent disease in the elderly”, “Health improvements from the implementation of strategies to promote health and prevent disease in the elderly”, and “The impact of the participation in these strategies to promote health and prevent disease in the elderly had on the formative process”. In the narratives of the students, several indicators emerged from the analogical and progressive classification of the elements, as expressed in Table 1.

To guarantee the quality of the investigation, several procedures were developed, among which we point out:i.Data collection and analysis procedures were explained, as well as the entire theoretical perspective, which was the framework for the study.ii.Participant review: The results of the analysis carried out by the main investigator were sent back to the participants so they could verify whether the interpretations actually reflected their ideas about the topic.iii.Peer/judge revision: Judges/experts on the subject were asked for collaboration, so they could validate the content analysis and make suggestions for improvement.

## 4. Results

This study involved ten students in the last year of the nursing graduation course. Their ages went from 22 to 28 years of age, with a mean of 23.8. Most (7) were female.

The analysis of the narratives of this group of students allowed us to analyze their perspectives about the practice of health promotion and disease prevention in the elderly.

We present the results based on the selected categories and the indicators identified in the content analysis.

### 4.1. Strategies

In this category, we present the perceptions of the students in regard to the different strategies to promote health and prevent disease in the elderly used in their educational process. The following indicators emerged in the narratives:

#### 4.1.1. Adequate Information and Communication

Adequate information and communication are determinants for the promotion of healthy lifestyles and the prevention of disease in the elderly. In the following registration unit, the information focuses on the promotion of healthy lifestyles.


*[…] I managed to provide information about the adequate diet people should follow, to encourage the practice of exercise, to encourage vaccination, to encourage them to reduce drinking and smoking (to elders with these habits). (P1)*


*Proper information and communication focus on creating a therapeutic relationship in this registration unit*.


*Some strategies were adopted to promote health and prevent disease in the elders, focusing on the creation of a therapeutic relationship, which is the development of a bond between the elder and the nursing worker; on the daily reality of the elder; on the adoption of active listening, availability, and empathy with the person, that is, on using an adequate and efficient communication. (P3)*


#### 4.1.2. Health Education

This strategy was the most mentioned one, both in regard to the number of students who mentioned it (9) and to the number of thematic units (14). It involved the elderly, in general, in encouraging the adoption of healthy lifestyles, people with diabetes and hypertension, and elderly people who attended diabetic foot consultations, as shown in the following registration units.


*The strategies selected included a session of education to elders who went to diabetic foot consultations in that health center. To do so, people were invited, a PowerPoint presentation about the topic was created and presented to those who were present in the session before it started. (P2)*



*[…] developing health education groups; elaborating programs targeted at the health of the elder. (P3)*



*Therefore, these surveillance consultations, specialized and/or general, make it possible for nurses to act in the fields of disease prevention, in people with higher risk for specific pathologies or for health promotion, through the encouragement of the adoption of healthy lifestyles, be it in those who have a specific disease, in those who may develop a certain disease, or in healthy people. (P4)*


Health education involved caregivers of the elderly during home visits, as shown in the following registration unit.


*I carried out health education sessions on the benefits of exercise; informative pamphlets/discourses during consultations about the importance of the flu vaccine; education in health for the caregivers, during home consultation, about pressure ulcer prevention in bedridden clients; informative discourses about the dangers of polypharmacy and of the non-adherence or incorrect administration of therapy; leaflets and informative discourses about the preventive measures against the dissemination of COVID-19; health education about measures to prevent against MRSA […]. (P4)*


Empowering the elderly to be independent in activities of daily living and adopting healthy lifestyles were the focus of health education in the following registration units.


*It should be mentioned that I was involved in diverse strategies, and it should be highlighted that most took place in the clinical classes in health centers, where I could carry out health education, provide information to elders on the adoption of healthy lifestyles that they should follow, including the prevention of the use of alcohol and tobacco, encouraging the adherence to vaccination, encouraging the practice of exercise, also being involved with the control of risk factors, such as overweight, arterial hypertension, diabetes mellitus, and dyslipidemia. (P5)*



*[…] intervention directed at promoters of healthy diets, which integrated information about the repercussion of bad dietary habits, the teaching of a balanced diet with no excessive salt, sugar, or fats, and instructions on the correct way to make foods, directed at the caregivers/institutions. (P6)*



*[…] projects of exercise, which implied in fomenting the importance of regularly exercising or walking in an open space and its benefits for health, in addition to physical activities in group, making available informative and illustrative leaflets. (P6)*


#### 4.1.3. Home Care

Home care involving the elder and their family was considered to be a health promotion and disease prevention strategy.


*The creation and organization of networks of care to be provided at home and at the outpatient clinic, giving support to the specific needs of the elder and their relatives, maintaining the wellbeing and comfort at home, a strategy important to avoid hospitalizations or institutionalization. (P6)*


#### 4.1.4. Social Projects

Social projects were mentioned by a student as a strategy used in the promotion of health and prevention of disease in the elder.


*[…] the development of social projects […] (P6)*


### 4.2. Health Benefits

From the perspective of these students, many benefits to their health came from their use of strategies to promote health and prevent disease in the elderly. The several benefits to health, expressed in the narratives of these students, show how they valued the rich strategies used in different contexts of care.

#### 4.2.1. Disease Prevention

The strategies used were determinant to prevent diseases and several complications, as the statements below indicate.


*[…] reduction of the number of elders with minor and major complications associated with diabetes, hypertension, obesity, etc. (P4)*



*[…] the diminution of COVID-19 infections. (P4)*



*[…] the diminution of complications from the disease, the stabilization of symptoms. (P6)*



*The health benefit was avoiding ulcers and many complications that could have emerged from this injury. (P9)*


#### 4.2.2. Health Promotion

Health promotion, as a health benefit, appears as associated with awareness about the disease.


*[…] increased awareness about the disease… P4*


#### 4.2.3. Wellbeing and Self-Care

Wellbeing and self-care were the indicator of this category, expressed by a high number of students (6). The frequency of thematic units (10) suggests the importance attributed to these health benefits that resulted from the strategies to promote health and prevent disease in the elderly in a clinical context.


*[…] which will consequently improve their quality of life. (P3)*



*The health benefits resulting from the implementation of these strategies to promote health and prevent disease in the elderly include increased global wellbeing, as well as improved health state and quality of life of the elder […]. (P5)*



*[…] weight loss, progressive increase in functional independence. (P6)*



*[…] the comfort and satisfaction of needs and family harmony. (P6)*



*Increased autonomy and independence of the elder as they carry out their daily life activities. (P8)*



*[…] thus allowing an increase in the quality of life. (P9)*



*[…] in addition to the improvement in their quality of life. (P10)*


#### 4.2.4. Behavioral Change

The strategies to promote health and prevent disease caused behavioral changes in the elders, particularly in regard to changes in risk behavior and the adoption of healthy lifestyles.


*[…] at the time of the presentation of the education session, the elders were interested and willing to change some risk behaviors they had to improve their health and prevent disease. (P2)*



*[…] the sessions led to many health benefits; specifically, the adoption of healthy lifestyles, the reduction of polypharmacy, the creation of a calm environment, adapted to each elder […] (P10)*


#### 4.2.5. Empowering

One of the students highlighted the empowering of the elder as a health benefit, which allows elders to be prepared to make decisions for the implementation of strategies connected to their health conditions (Vale et al., 2018).


*[…] the empowering of the elder. (P3)*


#### 4.2.6. Client Satisfaction

The satisfaction of the elder with the valorization and support provided is evident in the narratives of these students, reflecting on their activities in regard to the workers that intervene in health promotion strategies.


*When the elder feels valued and supported, they will be willing to actively listen to the things the professional has to say and teach in regard to the promotion of their health. (P3)*


#### 4.2.7. Management of the Therapeutic Regime

One of the benefits from the narratives of the students in regard to the strategies to promote health and prevent disease in the elderly was the management of the therapeutic regime.


*[…] correct adherence to the therapy […] (P4)*



*Increased adherence to the therapeutic regime. (P8)*


#### 4.2.8. Expenses

The reduced economic costs were also highlighted as a health benefit from the strategies to promote health and prevent disease in which these students participated in the clinical context.


*[…] the reduction of the costs inherent to the disease process […] (P3)*



*[…] will reduce the need to use health services and, therefore, reduce the economic needs related to health. (P5)*


#### 4.2.9. Life Expectancy

The increased life expectancy was also mentioned by a student as one of the health benefits resulting from the implementation of strategies to promote health and prevent disease.


*In regard to the male population, there have been some life expectancy increases, especially after 60 years old. […] Portugal has one of the lowest male life expectancies, and the health benefits are not high when compared to other European countries. (P7)*


### 4.3. Impact on the Formative Process

From the perspective of these students, participation in health promotion and disease prevention strategies impacted their learning, the development of competencies, and the awareness about these competencies.

#### 4.3.1. Learning and Developing Competencies

The learning and the development of competencies were indicators that emerged from the narratives of five students. The participation of these students in the strategies to promote health and prevent disease in the elderly helped them to understand the importance of education for health and to develop more efficient and effective health education interventions, as can be seen in the following registration units.


*[…] it had quite a positive impact on my formation process, as it helped me realize the importance of education in health and the effect it has on the health of people. In this regard, I could understand that if you intervene early, you can change certain risk behaviors and prevent disease, thus avoiding other, more serious issues. (P2)*



*My participation in the strategies to promote health and prevent disease in the elderly allows me to understand what are the most effective and efficient interventions, as well as to develop technical-scientific competencies in health that directly affect the behavior of the elderly. (P3)*


Building new practices and understanding the importance of empowerment of the elderly person are implicit in the learning and skill development of these students.


*It also allows for the construction of new practices of health, from the perspective of reflecting and adopting new methods to care for the elder, based on health promotion and disease prevention. (P3)*



*[…] academic education has an essential role, as it provides us with several qualities, such as creativity, resilience, pragmatism, perseverance, the capacity to deal with the unpredictable, to work and cooperate with other sectors of society, to use research and maximize their importance, etc.… which are paramount tools for us to become competent health professionals and the best facilitators for the promotion of a healthy and active aging, free from prejudice and social stigma. (P6)*


The students’ participation in these strategies helped them to acquire new skills and knowledge.


*My participation in these strategies allowed me to understand that when we encourage and try to strengthen the abilities of the elder, they feel their competency as individuals, valued, and are predisposed for more active participation. The elder feels empowered and capable of taking control of their own lives, thus being more receptive to participation in health care. (P8)*



*My participation in these strategies to promote health and prevent disease in the elderly allowed me to apprehend new competencies and knowledge, especially in regard to health promotion and disease prevention in the elder, enriching my formation process as it allowed me to carry out correct interventions in the future. (P10)*


#### 4.3.2. Raising Awareness of Competencies

Participating in these strategies with the elders was determinant to raise their awareness about their capabilities, especially in regard to training persons and health promotion, as manifested in the statement below.


*The fact that we are constantly directly or indirectly connected to the act of caring, instructing, educating, and intervening with the person makes us aware, as our academic formation advances, that, among all tasks a health professional has, in the specific case of the nurse, training and promoting health, which mostly involves behavioral changes, is the most complex one, because it requires continuous, aware, and desirable investment. (P6)*


## 5. Discussion

The results of this qualitative study reiterated that clinical classes, as a form of teaching, allow for an integration of theory and practice in learning [23], influencing the perception of students about health promotion and disease prevention in the elder. The connection between “school desk” knowledge and practice is vital to support the making of clinical decisions [24], and in this concrete case, for the learning of strategies that allow for the health promotion of the more vulnerable elder population. We corroborated the findings according to which elders with multiple morbidities need more complex health care, and health promotion is still not a priority for this group of people [25]. Aging is still one of the greatest challenges for health and social systems in general and for nursing care in particular. To respond to it with cost-effective solutions, effective strategies are needed that can guarantee a healthy and active life for those who are aging [26]. Evidence-based strategies will only be prescribed and implemented if health professionals have the necessary knowledge and develop competencies to lead this process. However, the results of the research indicate that one of the barriers to the dissemination of health promotion is the shortcoming in the preparation of future nurses [26]. This statement assumes that it is naive to presume that the inclusion of the topic on nursing courses syllabi will be enough to deal with current needs [26].

For future nurses to assume this role, they need to understand and have experience in it as students [26,27]. They must also learn to work in teams that integrate health promotion [24] as a core element of their intervention with the elderly.

As a result, the findings of this investigation allow one to perceive that clinical learning in the community, together with the elders and their families, and with learning results targeted at health promotion and disease prevention contribute to overcoming the aforementioned barrier. Therefore, this clinical learning strategy allows for the provision of care that is more focused on people and adjusted to their needs, expectations, and values, with better results in regard to therapeutic regime adherence [28].

In the category “Strategies to promote health and prevent disease in the elderly”, the students expressed how much importance they attribute to different strategies to promote health and prevent disease during their formation process. Home care, information, communication, and health education stood out. As other studies have observed, the implementation of health promotion in home care is important to reiterate the self-care abilities of the elder and increase or improve their knowledge about their own health, allowing them better opportunities to decide about their own care [24,29,30].

It should be highlighted that only one student mentioned social projects and their influence on health promotion. Social prescriptions are a recent topic in health and emerge in literature as an approach that promotes the use of non-clinical local activities, such as exercise, social, technological, and touristic activities, as recommended by health professionals. This could help improve one’s physical and psychological wellbeing, health behavior, and self-efficacy and reduce expenses [31], treating diseases and complications associated with the reduction of physical activities and social participation. Due to its potential, it should be included in the syllabi of health courses.

Regarding the category “Health benefits from the implementation of the strategies to promote health and prevent disease in the elderly”, the students valued, among other elements, wellbeing, self-care, behavioral changes, and empowerment. The transference of knowledge about health and healthy lifestyles to the elder, targeted at their individual needs and using cognitive-behavioral strategies adequate to their level of literacy, reiterates responsibility, promotes self-esteem and feelings of safety, and increases the probability of a person to remain independent at home for longer [24].

On the other hand, the promotion of self-care and the prevention of complications influence the management of the therapeutic regime and the adherence to treatment, also affecting expenses and quality of life, since interventions to promote health and prevent disease may help avoid urgency services, avoiding human suffering, the use of community resources, and probably reducing the risk for errors associated with health care [24], such as increased risks for infection and falls at the urgency services, or hospitalization.

The possibility of observing these “in loco” benefits reaffirms what the student has learned, making them conscious about the importance of these interventions for the quality of life of the elderly, including the management of their therapy, prevention, complications, and their own satisfaction.

The discourse the participants wrote allows one to observe that the articulation between theory and practice and the possibility of working with health promotion elements in the clinical context allows for the learning and/or development of competencies related to them, about which the students become more conscious. Opportunities to plan and implement health education sessions and interventions to promote health and prevent disease, as well as the confront between what was learned and what was implemented in the clinic, lead to the development of competencies related to communication and transmission of information, which is in accordance with aspects related to behavioral changes, which are paramount to improve adherence to the therapeutic regime.

Research with nursing students, aiming to analyze the knowledge of students from the last year of the nursing course about the concepts of health promotion and disease prevention, found that the students knew the concepts and that the relation between theory and practice is important [27] since most of them understand the concepts, but show difficulties to apply them in the daily practice of health services.

It is important to highlight the importance of nursing educators in this process. They can create a syllabus based on the principles of health promotion, which, later, are reflected in clinical experience [26]. The other aspect that should be taken into consideration is the use of information and communication technologies in the care of isolated elderly people in a pandemic context. Its use may be a solution to be considered, but it is important to prepare students for its application [32].

In this line of thought, we believe that the findings of this study have important implications for the teaching of nursing and for health and education policies, as they allow for the understanding of the strategies used in the promotion of health and prevention of disease in the elderly, in addition to the benefits obtained from the implementation of these strategies.

### Limitations

This research has limitations related to the method, the narratives, and the eligibility criteria of the participants, that is, with regard to sample size, the e-mail methodology applied in data collection, and the “intentional” selection of respondents. Further studies, with more participants, should delve deeper into strategies that allow for the development of competencies in the provision of care to elders, especially focusing on health promotion and disease prevention strategies in the context of the community.

## 6. Conclusions

Our findings allowed us to understand how students developed competencies in the promotion of health and prevention of diseases in the context of clinical classes. The analysis of the discourses led to the emergence of topics such as “Strategies to promote health and prevent disease in the elderly”, “Health improvements from the implementation of the strategies to promote health and prevent disease in the elderly”, and “The impact your participation in these strategies to promote health and prevent disease in the elderly had on your formative process”. These findings can contribute to the construction of syllabi in nursing graduation courses, focusing on strategies of health promotion and disease prevention that revolve around the needs and expectations of the elderly, aiming to improve their adherence to therapy.

## Figures and Tables

**Table 1 jpm-12-00306-t001:** Categories and indicators; Beja, 2021.

Category	Indicator
Strategies	− Adequate information and communication − Health education − Home care − Social projects
Health benefits	− Disease prevention − Health promotion − Wellbeing and self-care − Behavioral change − Empowering − Client satisfaction − Management of the therapeutic regime − Expenses − Life expectancy
Impact on the formative process	− Learning and/or development of competencies − Raising awareness of competencies

## Data Availability

Data are available only upon request to the authors.

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
