# Peer review of "Health Promotion and Disease Prevention in the Elderly: The Perspective of Nursing Students"

_jpm, 2022, doi:10.3390/jpm12020306_

Round 1

Reviewer 1 Report

Abtract: Should be structured.  In the abstract one introduction should be provided and in addition, some results should be presented.
Material and Methods:  Should be more structured. First of all the variables should be categorized. If the  data was collected using narratives, the process of codifying should be described
The analysis of the narratives is described in results, and it should be described in methods.
Discussion correct. 

Author Response

Abtract: Should be structured.  In the abstract one introduction should be provided and in addition, some results should be presented.

Thank you for the feedback, as suggested; Background was inserted in the abstract (page 1, line 19-21).

Material and Methods:  Should be more structured. First of all the variables should be categorized. If the  data was collected using narratives, the process of codifying should be described.

We thank you for your comments and regarding them, we would like to highlight the following points:

  1. The narratives received constituted the corpus of analysis, which corresponds to the material to be analyzed and produced for the research (Bardin, 2016);
  2. In the coding process, we defined and cut out units of record. The theme was defined as a unit of record, which, in line with what Bardin (2016) argues, is a statement about a subject, to which a wide range of singular formulations can be affected. We considered as unit of context, as unit of understanding to codify and understand the exact meaning of each unit of record, the answer of each participant to the guiding questions of the narrative. And these, given their characteristics, were fundamental to ensure the internal validity of the process..
  3. Categories were defined and the registration units were regrouped in the different categories. The indicators were also defined, corresponding to the inferences, as recommended by Bardin (2016), which were assumed to be logical deductions about the subject under study. In this process, the criterion of semantic categorization was used.

All these aspects are described in the methodology and support the presentation of the results.

The analysis of the narratives is described in results, and it should be described in methods.

We thank you for your suggestions, and although in other studies this has been done, we agree that the analysis grid should be the summary of the analysis process, and in this case, it can be presented at the end of the methodology. We have introduced this change in full agreement with your recommendations.

Discussion correct. 

Reviewer 2 Report

General comments

=============

The paper “Health Promotion and Disease Prevention in the Elderly: The perspective of nursing students” aims at understanding “how the experiences of nursing students in a clinical teaching context contributed for the development of their competences in the promotion of health and prevention of disease in the elderly.“ (see abstract). As a methodology, the study has used a qualitative approach by collecting narratives of n=10 nursing students via e-mail. As selection criteria for participation the students had to reveal “rich and diverse experiences in a clinical context of provision of health care to the community, focusing on the promotion of health and the prevention of disease in the elderly” (see. p. 3, lines 113-115).

This manuscript is well written and easy to understand, and it is obviously the result of a significant and commendable effort. The topic is especially important with regard to the overaged population of Portugal and in the face of a tremendously duplicating aging index, cited in the manuscript, between 2015 and 2080 from 147 to 317 elders per 100 young people. Thus, the topic is really up-to-date and although the database is a small one, this does not constitute a problem, because low numbers of respondents are characteristic for qualitative research (e.g., Vanderstoep & Johnson, 2008).

Although I have read the manuscript with great interest, I must admit that it is not entirely comprehensible to me, what the empirical study was targeting at. So, the core contribution should be stated much more clearly, and it should be fully stated, what the qualitative study sheds light on in detail.  When looking at the description of the sample recruitment, it is stated that the sample was “intentional (non-probabilistic), selected by the investigators” (see p. 3, lines 112-113). The specific sample composition, however, should be better justified in order to eliminate criticism with regard to a selectivity bias or a lack of objectiveness. In my opinion, the sample and specific sample does not discount the value of the gained insights, but nevertheless, results should be interpreted in the light of this specific sample description. The characteristics of the sample composition and the small number of respondents should be included in the Limitations section.

In my view, although I have read the manuscript with great interest, there are some weaknesses that should be addressed regarding the paper, as described in detail below. I believe that it should be possible to tackle them with a certain amount of effort. Please regard the following points as constructive criticism.

Specific comments

=============

Major comments

---------------------

  1. Please explain what considerations were regarded when selecting the non-probabilistic sample. What criteria played a role and how were they selected? Although the selection criteria of being a student in the 4th year of the nursing graduation course of the specific institution was mentioned and having been approved in the Nursing Clinical Classes in Community Health as well as having been involved in practices were crucial, it is not stated if all of the students of the 4th class were invited to take part in the survey or not. The authors should elaborate on this.
  2. The authors should explain how the guiding questions were developed and why instead of focus group discussions or personal interviews a survey via e-mail was sent out. This is not reasonable to me. Usually, respondents are not willing to take much time and effort to write a lot and respondents cut down the burden of writing and do not write a lot in open-ended questions. So, did the respondents receive any incentives? Were they guaranteed anonymity? Why did the authors opt for an e-mail-survey instead of semi-structured interviews or something of the kind? The three guiding questions are very diffuse in my view, and I wonder that respondents knew what to answer. How did the authors ensure comprehension of the questions?
  3. Although the Introduction is concise, I would recommend that the authors think about elaborating the theoretical background immediately after the concise introduction.
  4. The research questions are a little bit vague, and the authors might think of some theories that could be useful. In its present form, although the importance of the topic was made totally clear, the derivation of the research aim was not clear to me at all. Maybe it would help to give a literature review in the field of health promotion and disease prevention for the specific target group of elderly and the role of nurses in this respect. I must admit that my own research background is very far away from this topic, but nevertheless I would classify the study as important, having the tendency to an overaged population for many European countries in mind. Maybe it is only the core contribution which is not totally clear to me, and I would invite the authors cordially to make their core contribution much clearer.
  5. Additionally, the literature base is expandable. Although a study can be exploratory in nature, a solid theoretical framework should never be neglected. Of course, the authors can refer to some of the references when interpreting the results in the discussion section, but authors should not neglect them in the theoretical background at the beginning. There should not be a popping-up of many central references in the discussion section, without mentioning them earlier, because they are interesting and quite useful for providing a theoretical framework at an earlier stage. In my opinion, the authors would be well advised to include a thorough appraisal of the existing literature immediately after the introduction. There is plenty of literature on nurses and also with regard to nursing education e.g., recent studies dealing with the challenges caused by the COVID 19-pandemic like the paper by Woo et al. (2021). All of them used qualitative interviews and none of them used e-mails. In any case, a profound theoretical framework would enrich the meaningfulness of the results significantly. The authors are cordially invited to look at these papers with regard to the methodology applied and how they described the results.
  6. The entire Results section seems odd to me. In my view, it is very unusual to schedule an entire list of verbatim quotes one after another without interpreting, comparing or interrelating the quotes with each other. Especially on pages 6-9, there are far too many verbatim quotes. Maybe you could look at the paper by Calma et al. (2021) as an example for a scientific paper using a qualitative approach or the paper by Woo et al. (2021). Additionally, a guidance on the usage of quotations was developed in the paper by Eldh (2020). The latter delivers a critical examination of the application of quotations in the reporting of qualitative studies. The authors should read the papers carefully and try to revise their Results section noticeably afterwards. Usually, each subtheme is demonstrated by one single verbatim quote and not a schedule of verbatim quotes listed one after the other without interrelating them like it was done in the present manuscript. The authors should revise their manuscript substantially in this respect.
  7. The section describing the procedure applying the analysis of the narrative material is very short and diffuse. Why did the authors not deploy the procedure of inductive category development by Mayring (2000) or Thematic Analysis by Braun and Clarke (2006), which are both widely used methods of qualitative data analyses? Usually, NVIVO or MXQDA are used for analysing qualitative data. I have never heard about “thematic category analysis”, which the authors opted for. The authors should clarify why they have chosen the method they applied.
  8. The authors should report their results by, e.g., by filling out something like the 32-item checklist for interviews and focus groups like the Consolidated criteria for reporting qualitative research (COREQ) (Tong, Sainsbury & Craig 2007)? Although the authors have applied e-mails for data collection, they nevertheless should take a look at the COREQ-guideline and justify what they have done. Not all of the items, but many of the COREQ-guidelines could be answered.
  9. The Limitations section should be amended with regard to the sample size, the e-mail methodology applied and the “intentional” selection of respondents.

Minor comments

---------------------

  1. The denomination of the three categories found is too complex and too long. The authors should think about cutting the denominations down and making them more concise.
  2. Abbreviations used in the main text should be explained in full the first time they are used, like e.g. “USF” and “UCS” (see p. 2, line 86). Did you mean OECD-countries instead of OCDE countries (see p.2, line 55)?
  3. The object of the study was included three times, thereof two times in the main text (see p. 3, lines 103-105, p. 2, lines 92-94). Nevertheless, the contents of the object still remains unclear (see my comments above).

Good luck with your research!

References:

Braun, V., & Clarke, V. (2006). Using thematic analysis in psychology. Qualitative research in psychology, 3(2), 77-101.

Calma, K. R. B., Halcomb, E., Williams, A., & McInnes, S. (2021). Final‐year undergraduate nursing students’ perceptions of general practice nursing: A qualitative study. Journal of Clinical Nursing, 30(7-8), 1144-1153.

Eldh, A. C., Årestedt, L., & Berterö, C. (2020). Quotations in qualitative studies: reflections on constituents, custom, and purpose. International Journal of Qualitative Methods, 19, 1609406920969268.

Tong, A., Sainsbury, P., & Craig, J. (2007). Consolidated criteria for reporting qualitative research (COREQ): a 32-item checklist for interviews and focus groups. International Journal for Quality in Health Care, 19(6), 349-357.

Vanderstoep, S. W., & Johnson, D. D. (2008). Research methods for everyday life: Blending qualitative and quantitative approaches (Vol. 32). John Wiley & Sons.

Woo, B. F., Poon, S. N., Tam, W. W., & Zhou, W. (2021). The impact of COVID19 on advanced practice nursing education and practice: A qualitative study. International Nursing Review, 1–10. (retrieved from: https://doi.org/10.1111/inr.12732, 01-24-2022)

Author Response

Dear reviewer:

Thank you for your careful and reasoned review of our article. I attach the point by point response to your recommendations.

Best regards;

Round 2

Reviewer 1 Report

I think it can be accepted in the present form  if the abstract could be improve givins a short introduction. 

Author Response

Dear reviewer:

Thank you very much for the suggestion we have integrated by putting a paragraph with introduction in the abstract (page 1, line 19-22). 

Reviewer 2 Report

Dear authors!

Thank you for submitting a revised version of the manuscript with the title "Health Promotion and Disease Prevention in the Elderly: The perspective of nursing students."

I appreciate the changes made, and, in my opinion, the quality of the manuscript has improved significantly. You have addressed nearly all of my comments in a satisfactory way. To sum up, I think that the manuscript has improved through the revision, and it could make a nice contribution to the literature. From my point of view, there remain only two major and one minor concerns that should still be smoothed out before publication. Altogether, I think the manuscript will be publishable after addressing these remaining points satisfactorily. It should not be a problem for you to quickly smooth out the remaining issues. I will recommend the publication of the manuscript to the editor immediately after you have considered the following major and minor comments.

=============

Major comments

---------------------

  1. I also appreciate amending the Results section. However, there are still too many quotes in my view, and thus, the Results section is still challenging to read as a large number of quotes and thereby indenting the main text disturbs the reading flow.
  2. I appreciate the new Background section inserted and having expanded the literature base. Well done! I agree that it helps in understanding the rationale for the empirical study. In the introductory passage of my former review, I proposed the following: "Although I have read the manuscript with great interest, I must admit that it is not entirely understandable to me, what the empirical study was targeting at. So, the core contribution should be stated much more clearly, and it should be fully stated, what the qualitative study sheds light on in detail." However, in my view, this point has not been appropriately addressed. Also, in the revised form of the manuscript, the statement of a scientific aim and specific research questions have been missing. What was your goal? It would help if you tried to cut it down to one sentence rephrasing the aim of the empirical study. The following objective included is not clear from a scientific point of view: "This study aims to understand how the experiences of nursing students in a clinical teaching context contributed for the development of their competences in the promotion of health and prevention of disease in the elderly." Maybe you should try to rephrase it to make it understandable for the average readership of JPM. Perhaps, the first step would be to put it into a grammatically correct form: "This study aims to understand how nursing students' experiences in a clinical teaching context contributed to the development of their competencies in the promotion of health and prevention of disease in the elderly." Maybe adding one or two sentences about the usefulness of these insights could be given? Why is it helpful to know that experiences lead to competencies? The causal chain of experiences leading to knowledge is entirely logical and nothing unexpected. So why is it nevertheless worthy of being investigated? What is the added value of this knowledge? What can be done with this knowledge? These questions should be answered and shed light on your manuscript's core contribution.

=============

Minor comments

---------------------

  1. With regard to my comment no, I asked the following: "Why did the authors not deploy the procedure of inductive category development by Mayring (2000) or Thematic Analysis by Braun and Clarke (2006), which are both widely used methods of qualitative data analyses? Usually, NVIVO or MXQDA are used for analyzing qualitative data. I have never heard about "thematic category analysis," which the authors opted for. The authors should clarify why they have chosen the method they applied." You mentioned in your answer that the procedure you applied is "the most widely used technique of qualitative content analysis." and you refer to a publication by Bardin (2016). This source is not available in English. Maybe this technique is widely used in Portugal or Brasil, but I doubt it is well known in the rest of the world. This, however, does not constitute a severe problem in my view. Thanks for revealing your opinion in this respect. Maybe you should narrow down the claim of "the most widely used qualitative content analysis technique " or provide evidence for this claim.

All of the other issues raised have been appropriately addressed.

Good luck with your research!

Author Response

Dear reviewer:

Once again we thank you for your careful review of our article. In this revision we have tried to respond to the requests made.

We have removed some quotes to make the reading of the article more fluid, trying to keep the quotes that best translate the analysis carried out. 

We have reworded the objective and introduced one more paragraph in an attempt to respond to the request (page 3 – lines 105-114).

Regarding the minor comment - Laurence Bardin is a French methodologist, which in 1977 published the 1st edition of the book 'L' Analyse de Contenu' influenced many social and health researchers in France and quickly spread throughout Europe and Brazil, with several updated editions of the proposed method. In Portugal it is widely used in education and health. The authors that you mention are unknown to us, but we are interested in exploring their work and their proposal for the content analysis technique.

We did not use software for the qualitative analysis of the findings.